# Effects of Vegetation Types and Soil Properties on Regional Soil Carbon and Nitrogen in Salinized Reservoir Wetland, Northeast China

**DOI:** 10.3390/plants12213767

**Published:** 2023-11-03

**Authors:** Yuchen Wang, Heng Bao, David J. Kavana, Yuncong Li, Xiaoyu Li, Linlu Yan, Wenjing Xu, Bing Yu

**Affiliations:** 1College of Wildlife and Protected Area, Northeast Forestry University, Harbin 150040, China; yuchenw@nefu.edu.cn (Y.W.); baoheng1990@163.com (H.B.); david.kavana@sua.ac.tz (D.J.K.); yll1614266763@nefu.edu.cn (L.Y.); xwj7297@nefu.edu.cn (W.X.); 2Department of Soil and Water Sciences, Tropical Research and Education Center, IFAS, University of Florida, Homestead, FL 33031, USA; yunli@ufl.edu; 3School of Fishery, Zhejiang Ocean University, Zhoushan 316022, China; lixiaoyunefu@163.com

**Keywords:** microbial biomass, carbon and nitrogen, salinized soil, vegetation type, reservoir wetland

## Abstract

This study investigated the spatial variability in soil organic carbon (SOC), total nitrogen (TN), soil microbial biomass carbon (SMBC), soil microbial biomass nitrogen (SMBN) and their possible relationships with other soil properties in the Hongqipao reservoir, which is dominated by different vegetation types. The results showed that there were high spatial variabilities in SOC, TN, SMBC and SMBN, and that the SOC, TN, SMBC and SMBN contents decreased with increasing soil depth in the Hongqipao reservoir. The SOC was significantly positively correlated with TN, SMBC, moisture content (MC) and negatively correlated with carbon to nitrogen ratio (C:N ratio) and bulk density (BD). Soil TN was significantly positively correlated with SMBC, SMBN, MC and negatively correlated with the C:N ratio, BD and pH. The SMBC was significantly positively correlated with SMBN, MC and negatively correlated with the C:N ratio, BD and pH. The SMBN was significantly negatively correlated with the C:N ratio and BD. All of the measures of soil properties in this study could explain the higher significant variability in the response variables (SOC, TN, SMBC and SMBN contents). The generalized additive model (GAM) showed that SOC and TN had different influencing factors in different soil depths. The structural equation model (SEM) showed that vegetation types had a significantly positive effect on TN and SMBN, and the soil depths had a significantly positive effect on SOC and a significantly negative effect on TN and SMBC. This study further suggests that vegetation types play a major role in determining the spatial characteristics of soil carbon and nitrogen, and any changes in the vegetation types in the reservoir may influence the distributions of soil carbon and nitrogen. This may affect the global carbon and nitrogen budgets and the atmospheric greenhouse gas concentration significantly.

## 1. Introduction

Soil carbon and nitrogen are essential elements of sustainable soil fertility and productivity and can substantially affect climate change through carbon and nitrogen emissions (including carbon dioxide (CO_2_), methane (CH_4_) and nitrous oxide (N_2_O)) [1,2,3]. As one important pool of soil organic matter in wetland ecosystems, wetland soil serves as the source, sink and transfer of nutrient and chemical pollutants [4,5]. It is estimated that 202–377 Gt carbon is stored in the top 100 cm of wetland soil since wetland provides the optimum environmental condition for the sequestration and long-term storage of carbon [6,7]. Reservoirs are an important type of wetland, where their combined area across the world has increased in recent years, and is now occupying approximately 5 × 10^5^ km^2^, which is 1/3 the size of all natural lakes [8]. Since reservoirs have a major impact on the biogeochemical cycle by acting as a storage of SOC on the local and global scale, several researchers have insisted that reservoirs require more scientific studies [2,9,10]. However, limited studies have been carried out to determine the role of reservoirs in the storage of SOC and nitrogen [2,11].

Studies have revealed that the spatial variation, accumulation and distribution characteristics of SOC and nitrogen in wetland soil are influenced by vegetation types, hydrology (water level fluctuation), soil microbial community, pH, salinity, temperature, soil moisture and many other factors [12,13]. Change in the hydrological regime can have a substantial effect on the soil properties, particularly carbon and nitrogen accumulations and releases, due to alterations in their chemical forms and spatial movements [14,15]. The waterfront zones of wetland have been reported to have abundant vegetation and soil microbes, as well as a higher capacity for carbon deposition than other land types [16]. Different types of vegetation communities and their development have an influence on SOC content and soil primary productivity [17]. Soil mechanical composition, BD, salinity, and nutritional status influence the capacity of vegetation directly and affect the input and output of soil carbon [18]. Moreover, vegetation growth affects the SOC content and distribution by changing the surrounding environment, especially the rhizosphere microenvironment, such as soil salinity, pH value, water, etc. [19].

Studies on the spatial variabilities of SOC and TN have been a popular topic in soil ecology in recent decades [1,3,20]. For example, the spatial patterns of SOC and TN within 0–80 cm for mine soil in the Loess Plateau area and the spatial variabilities of soil carbon, nitrogen, phosphorus contents and storages in an alpine wetland in the Qinghai–Tibet Plateau have been studied [3,21]. More studies on the spatial variability in soil properties have been conducted in the southeast region of Xi’an in Shanxi province [22] and the Three Gorges Reservoir area, China [11]. However, similar studies are lacking on the Hongqipao reservoir in northeast China, despite the fact that this reservoir is characterized by different vegetation types which create high environment heterogeneity. The objectives of this study were (i) to determine the spatial distribution characteristics of SOC, TN, SMBC and SMBN in the Hongqipao reservoir; (ii) to assess the effect of environmental factors on SOC, TN, SMBC and SMBN in the Hongqipao reservoir and (iii) to evaluate the total SOC, TN, SMBC and SMBN in the Hongqipao reservoir.

## 2. Results

### 2.1. Spatial Variations in SOC, TN, SMBC, SMBN and Other Soil Properties

The spatial characteristics of soil physicochemical properties from the seven kinds of sub-plot are presented in Table 1 and Figure 1. In all of the sub-plots, the SOC decreased with increasing soil depth. One-way ANOVA revealed that the overall mean values of SOC differed significantly among all of the sub-plots (F (6, 28) = 11.076, *p* < 0.05), with the highest value of 32.46 ± 4.81 g kg^−1^ recorded in sub-plot D and the lowest value of 17.25 ± 4.31 g kg^−1^ measured in sub-plot B (Table 1). Like the SOC, the mean TN also differed significantly among the sub-plots (F (6, 28) = 4.443, *p* < 0.05). The mean soil TN from sub-plot D was the highest with 1.40 ± 0.49 g kg^−1^, while that from sub-plot B was the lowest with 0.48 ± 0.37 g kg^−1^. Moreover, soil TN varied with remarkably higher mean values being recorded in the 0–30 soil depth (Figure 1). 

The soil microbial biomass (SMB) which is an important parameter of nutrient cycling in an ecosystem also revealed spatial variation in this study. One-way ANOVA showed that both SMBC and SMBN weren’t significantly different in most subplots (Table 1). The mean SMBC in sub-plot D was the highest with 1156.67 ± 453.39 mg kg^−1^. The overall mean SMBN values in sub-plots D and F were 46.22 ± 78.82 and 47.97 ± 22.97 mg kg^−1^ respectively, which were significantly higher than those in other sub-plots. Unlike SOC and TN, which showed clear decreasing trends with soil depth in most sub-plots, there were no clear patterns of SMBC and SMBN with soil depth (Figure 1). The SMBC values were lower in the upper soil layers in both sub-plots A and B, while higher SMBN values were observed in the 0–10 cm soil depth in sub-plot D.

Soil BD increased with increasing soil depth in all of the sub-plots. One-way ANOVA revealed a significant difference in BD among the sub-plots (F (6, 28) = 5.166, *p* < 0.001). The highest overall mean BD was recorded in sub-plot B (1.58 ± 0.17 g cm^−3^) and the lowest was measured in sub-plots G (1.21 ± 0.07 g cm^−3^). The overall mean values of MC also differed significantly among the sub-plots (F (6, 28) = 15.252, *p* < 0.001), while that of soil pH did not differ (F (6, 28) = 1.991, *p* > 0.05).

### 2.2. Correlation Analysis between Soil Physicochemical Properties

The correlation coefficients between SOC, TN, SMBC, SMBN and other soil properties are listed in Table 2. The SOC was significantly positively correlated with TN, SMBC, MC and negatively correlated with the C:N ratio and BD. Soil TN was significantly positively correlated with SMBC, SMBN, MC and negatively correlated with the C:N ratio, BD and pH. The SMBC was significantly positively correlated with SMBN, MC and negatively correlated with the C:N ratio, BD and pH. The SMBN was significantly negatively correlated with the C:N ratio and BD. The C:N ratio was significantly positively correlated with BD and negatively correlated with MC. Soil MC was significantly negatively correlated with BD.

### 2.3. Relationships between SOC, TN, SMBC, SMBN and Other Soil Properties

The RDA was carried out to determine the relationships between SOC, TN, SMBC, SMBN and other soil properties measured in this study. From the RDA biplot (Figure 2), soil MC had a significantly positive influence on SMBC, SOC and TN. Soil BD and the C:N ratio had significantly negative influences on SMBC, SMBN, SOC and TN. The RDA results further revealed that pH had relatively small influence on SOC, TN, SMBC and SMBN. 

A multiple regression analysis was performed to model the relationships among SOC, TN, SMBC, SMBN and other soil properties, and the results of the regression analysis are presented in Table 3. For every response variable, two models are presented. One model included only those soil variables which were significantly correlated with the response variable under the Pearson correlation analysis (Table 2), and the second model included all of the soil variables. From the results (Table 3), 86.4% variability in SOC could be explained by the correlated variables. However, 86.3% variability in SOC was explained by all of the factors. The TN was correlated with all soil factors, so 91.4% variability in soil TN could be explained by all of the factors. Only 39.7% variability in SMBC could be explained by the correlated variables, while 44.1% variability in SMBC could be explained when all factors were included in the model. 40.2% variability in SMBN could be explained by the correlated variables, while only 36.8% of it could be explained by all of the variables.

### 2.4. The GAM for Analyzing Correlations between SOC, TN and Other Soil Physicochemical Properties in Different Soil Depths

The GAM was used to fit the changes in SOC and TN with other soil physicochemical properties in the 0–50 cm soil layer (Figure 3). In the 0–20 cm soil profiles, SOC was linearly positively correlated with TN (*p* < 0.001), and in the 20–50 cm soil layers, SOC was nonlinearly positively correlated with TN (*p* < 0.001), and it was also nonlinearly positively correlated with the C:N ratio in the 40–50 cm soil profile (*p* < 0.001). In the 0–50 cm soil profiles, TN had nonlinear correlations with the C:N ratio (*p* < 0.001).

### 2.5. The Effects of Vegetation Type and Soil Depth on SOC, TN, SMBC and SMBN

According to the SEM, the soil depth had a significantly direct effect on SMBC (path coefficient = −0.47, *p* = 0.002), SOC (path coefficient = 0.21, *p* = 0.002) and TN (path coefficient = −0.49, *p* < 0.001), but it had no significant correlation with SMBN. The vegetation type had a significantly direct effect on SMBN (path coefficient = 0.31, *p* = 0.016) and TN (path coefficient = −0.39, *p* = 0.004), but it had no significant correlation with SMBC and SOC. Moreover, SMBC had a significantly direct effect on SMBN with their path coefficients of 0.53, and TN had a significantly direct effect on SOC (path coefficient = 0.97, *p* < 0.001) (Figure 4).

### 2.6. Total SOC, TN, SMBC and SMBN Storages

In the 0–50 cm depth, TSOC ranged from 12.63 to 20.17 kg C m^−2^ across the soils dominated by different vegetation types. The TSOC differed significantly among the sub-plots (*p* < 0.05) (Table 4), and vegetation that was inundated had a relatively higher TSOC value than the dry vegetation. Sub-plot D had the highest TSOC, while sub-plot A had the highest TSMBC. Sub-plot D also had the highest TSN, but sub-plot F had the highest TSMBN. 

## 3. Discussion

### 3.1. The Spatial Variations in Soil Properties in Different Vegetation Type and Soil Depth

Studies on soil carbon and nitrogen in terrestrial ecosystems have received much attention around the world in recent years because the emissions of carbon and nitrogen oxides into the atmosphere have played a critical role in driving climate change [23]. Studies have illustrated that the spatial variations in SOC, TN and SMB are related to litter input, vegetation distribution, water-level fluctuation, soil moisture, temperature and nutrient condition [24,25]. The differences between organic matter input and output determine the amounts of soil organic matter and carbon [26]. In this study, spatial variations in SOC, TN, SMBC, SMBN and other soil properties were determined in the Hongqipao reservoir. The results showed that SOC and TN decreased with the increase in soil depth, which reflected the variations in the growth, distribution and decomposition of roots with soil depth. The vertical distribution of SOC also reflected that the plant detritus decreased with the increase in soil depth. When plant detritus decompose, most of their organic matters are mineralized and a new soil layer which contains a greater amount of organic carbon is formed [26,27]. 

Our study also revealed horizontal variations in SOC and TN in the Hongqipao reservoir. Sub-plots C, D and E had remarkably higher SOC and TN than other sub-plots. The reasons for this observation could be that, firstly, the high SOC and TN in the soils from these sub-plots could probably be explained by the additional organic matter supplied by the vegetation through litter falling associated with their relatively high productivity. Litters fell from *Polygonum amphibium* L., which was a dominant plant in sub-plots C and D, and it has been documented in the literature that it has a high concentration of carbon, magnesium and phosphorus [28]. Through determining the storages of organic carbon, nitrogen and phosphorus in the soil–plant system of *Phragmites australis* in the eutrophicated Mediterranean marsh, the litter falls of *Phragmites australis* were important reservoirs of organic carbon which were mineralized into the soil [29]. The accumulation and turnover rate of SOC tend to vary with vegetation type [30,31]. Secondly, since sub-plots A, F and G were located on dry places within the waterfront zone of the reservoir, their relatively low SOC concentrations could be because mineralization was favored in dry soil. Some studies had shown that hydrological processes (such as groundwater level, water depth, duration and frequency of flooding) affect the decomposition of litter; hence, hydrological processes could change SOC and TN [32]. 

The SMB plays an important role in SOC mineralization and nutrient cycling [33,34]. The compositions of vegetation species and soil water availability in wetland greatly influence the SMB [35,36]. In this study, a relatively high mean SMBC was observed in sub-plot D and E, which was mainly dominated by *Scirpus fluviatilis, Polygonum amphibium* L., *Carex tristachya and Phragmites australis*. A previous study also found that the vigorous root system of the Phragmites community increased the level of soil microbial activity which led to the increasing of its biomass, and higher organic matter from the Phragmites community increased soil carbon enrichment, which then helped to maintain soil microbial activities [37,38]. But the overall mean SMBN was relatively higher at sub-plot D and F. 

Although there were no clearly vertical profile patterns of the SMBC and SMBN, relative high values were observed in the top soil profiles (0–10 and 10–20 cm). This could be explained by the fact that the upper soil profile contained higher SOC and TN. Through investigating the responses of SOC and TN mineralization to the changes in SMB, SOC, TN and phosphorus stoichiometry, an increase in SOC and nitrogen accumulation resulted in higher SMBC and SMBN [33]. The mineralization of soil organic matter which mostly happens on the top soil profile (0–20 cm), as observed in this study, and most crucial fractions of SOC and nitrogen become parts of the microbe, such as phospholipid and protein in the process of mineralization, which are then released upon microbial turnover [39]. Correspondingly, the relatively high SMBC and SMBN in the top soil profiles indicate higher microbial metabolic activities in these profiles.

The BD increased with the increase in soil depth. The upper soil profile (0–10 cm) in all of the sub-plots had the lowest mean values of the BD compared to the other soil profiles. This is due to the fact that soil BD is considered to be the basic property that varies with the soil structural conditions and increases with soil depth due to changes in porosity, soil texture and organic matter content [40]. The mean BD among sub-plots differed significantly, with sub-plots A and B having remarkably high values. This could have been likely due to mineral sedimentation since these sites were located close to reservoir shoreline where the inflowing rivers pour water into the reservoir. A previous study found relatively high BD over a depth of 24 cm from a site located near the influent of a river flowing in a freshwater marsh [41]. The authors attributed their findings to the rate of mineral sediment accumulation, which tended to increase toward the lake and open water in a wetland [42]. Another possible reason for the relatively high soil BD in sub-plots A and B could be due to sediment settling out of the water column which was transformed into a soil layer, and then the soil layer bonded with the preexisting surface, thus decreasing the pore space and resulting in increased BD [43].

### 3.2. The Relationships between SOC, TN, SMBC, SMBN and Other Soil Properties 

The carbon cycle is closely linked with the nitrogen cycle through production and decomposition [44]. Our study also revealed that the soil TN concentration was significantly positively related with the SOC concentration in the 0–50 cm soil layer of the different vegetation types (Table 2 and Figure 2). This observation could be because the main nitrogen sources were the vegetation organic matter, litter and biological nitrogen fixation [45]. The Pearson correlation analysis and RDA also showed that soil MC was positively correlated with SOC and TN, and this concurred with the findings of other studies [46,47]. Under high soil MC conditions, the anoxic decomposition of soil organic matter tends to be inhibited, which results in the accumulation of SOC. Importantly, soil MC also affects nitrogen, since higher soil MC can hinder soil microbial activity; thus, it can create an environment that is not conducive to the mineralization and decomposition of soil organic nitrogen [48]. Therefore, this implies that a high soil MC can lead to a high soil SOC and TN concentrations.

Unlike soil MC which was significantly positively correlated with SOC and TN, the soil BD was significantly negatively correlated with the SOC and TN concentrations in the 0–50 cm soil profile (Table 2 and Figure 2). Through assessing stock and detecting the threshold of SOC and nitrogen along an altitude gradient in an east Africa mountain ecosystem, a previous study found that BD was negatively correlated with SOC and TN [49]. This observation could be likely due to the fact that the mineralization of SOC and nitrification of nitrogen were suppressed in soil with a high BD value [44,50]. It was documented in the literature that soil with low BD could store more SOC and TN because SOC and TN can be mobilized in porous spaces within the soil matrix [51]. The effect of soil properties on SOC and TN was also different in different soil depths. In the top soil, SOC and TN had great influences on each other due to the decomposition of a large amount of litter and plant debris. With the increase in depth, the humus in the soil decreased, and various soil properties had an impact on SOC and TN (Figure 3). This was because the vertical distributions of SOC and TN were influenced by vegetation characteristics and soil intrinsic factors (BD, MC, etc.). 

The SMB plays an important role in maintaining the soil structure, facilitating the microbial metabolic process and regulating the release of nutrients. When studying the effects of heavy metal and soil physicochemical properties on SMB and the bacterial community structure in wetland, a previous study found that there was a strong relationship between the soil physicochemical properties and SMB [52]. In this study, Pearson’s correlation analysis showed that SMBC and SMBN were significantly related with many soil properties, including SOC and TN. This could be attributed to the quality of SOC and TN [53]. 

From the multiple regression model results in this study, it was quite clear that all of the measured soil variables played a major role in influencing SOC, TN, SMBC and SMBN. A comparison of two models for each response variable of SOC, TN, SMBC and SMBN revealed a slight difference. One model included all of the soil variables measured and the other one included only those variables which were significantly correlated with the response variable. The SOC and SMBN could be explained better when the correlated variables were included in the model. But SMBC could be explained better when all of the variables were included in the model. This implied that all of the soil variables could play a role in influencing the SMBC when multiple factors were considered. 

## 4. Materials and Methods

### 4.1. Study Area

The study was conducted in the waterfront zone of the Hongqipao reservoir located in Nenjiang plain between Anda and Daqing city in Heilongjiang province (Figure 5). The size of the reservoir is 35 km^2^ with a capacity of 116,000,000 m^3^. This study area is under the influence of a mid-temperate continental monsoon climate characterized by a long and cold winter and high-temperature and rainy summer. The mean annual temperature in this area is 3.3 °C, with the lowest and highest temperature values measured in January and July of −37.2 °C and 38.3 °C, respectively. Moreover, the average precipitation received in this study area is 426 mm per year. During winter, the maximum depth of frozen soil is 2.14 m. The reservoir is the source of water to the people living in Daqing city. Ecologically, the Hongqipao reservoir provides habitats for about 40 species of fishes and rare birds. This reservoir is characterized by different species of plants, including *Phragmites australis*, *Typha angustifolia*, *Scirpus tabernaemontani*, *Potamogeton pectinatus*, *Potamogeton crispus*, *Spirodela polyrhiza*, *Scirpus yagara* and *Echinochloa crusgalli*.

### 4.2. Soil Sampling and Laboratory Analysis

Soil samples were collected from five sample plots (Figure 5). Seven kinds of sub-plots (coded as A–G) were selected based on their dominant vegetation species and hydrology conditions (Table 5), and three replicates were set for each kind of sub-plot in these five sample plots. Three sampling sites were set in one sub-plot, and the soil profiles for each sampling site were sectioned into five depths at 10 cm intervals (0–10, 10–20, 20–30, 30–40 and 40–50 cm); then, the three soil samples from the same soil layers were mixed to form a composite sample. Moreover, soil samples for the analysis of soil BD and MC were collected at each sampling site. The collected soil samples were portioned into two parts for every soil layer. One part of the sample for each soil layer was stored at 4 °C in sealed plastic bags to limit microorganism activity for the determination of SMBC and SMBN. The second portion of the sample of each soil layer was air-dried at room temperature in the laboratory for three weeks, and stones, roots and coarse debris were removed. The dried soil samples were ground to fine powder using a mill (FW100, Taisite in Tianjin, China), and then sieved through a 100-mesh sieve. The sieved soil samples were treated with 2 M HCl for 24 h at room temperature to remove carbonates [54,55]. The soil was then washed to pH > 5 with distilled water and dried at 40 °C. The SOC and TN were determined using an automatic elemental analyzer (Flash EA 1112, Milan, Italy). The soil pH was measured using a digital pH meter in the supernatant of a 1:5 soil/water mixture. The soil BD and MC were calculated on a dry weight basis. Soil MC was determined via the oven drying method (weighing before and after drying at 105 °C for 24 h). Soil BD was calculated by dividing the total dry weight of the soil sample by the volume of the core [56]. All results are expressed on a dry gram basis.

The SMBC and SMBN were determined using the chloroform fumigation extraction method [57,58,59]. The extraction was performed using K_2_SO_4_ (0.5 mol L^−1^) on unfumigated samples and fumigated samples in alcohol-free chloroform. The organic carbon and TN in fumigated and unfumigated extracts were measured using a Multi N/C 3100 SOC/TN analyzer (Analytik, Jena, Germany). The SMBC and SMBN were calculated as the concentrations of organic carbon and total nitrogen in fumigated samples subtracted by those in unfumigated (control) samples, with a conversion factor of 0.45 for microbial carbon and 0.54 for microbial nitrogen [60].

### 4.3. Determination of Aboveground Vegetation Biomass 

Three quadrants (1 m × 1 m) were established in each sub-plot. The vegetation in the quadrants was manually cut close to the soil surface and harvested and weighed immediately while fresh. The weighed samples were then taken to the laboratory and oven-dried at 80 °C to a constant weight. The dry biomass was calculated by multiplying the fresh weight of the harvested vegetation and the dry/wet ratios of the samples. 

### 4.4. Calculations of Total Carbon and Nitrogen

Total SOC (TSOC; g C m^−2^), total TN (TSN, g N m^−2^), total SMBC (TSMBC; mg C m^−2^) and total SMBN (TSMBN; mg C m^−2^) on a ground-area basis to a 50 cm depth were calculated according to the following formulas [23]:(1)TSOC=∑Di×Pi×SOCi×S
(2)TSN=∑Di×Pi×TNi×S
(3)TSMBC=∑Di×Pi×SMBCi×S
(4)TSMBN=∑Di×Pi×SMBNi×S
where Di, Pi, SOCi, TNi, SMBCi, SMBNi and S represent the soil thickness (cm), BD (g cm^−3^), SOC (g kg^−1^), TN (g kg^−1^), SMBC (mg kg^−1^), SMBN (mg kg^−1^) and cross-sectional area (cm^−2^) of the ith layer (i = 1, 2, 3, 4 and 5).

### 4.5. Statistical Analysis

Data were statistically analyzed using SPSS (version 19.0) and R 4.1.3 software. A one-way analysis of variance (ANOVA) was conducted to test the spatial variation in SOC, TN, SMBC, SMBN and other soil properties. Pearson’s correlation analysis and RDA were carried out to evaluate the relationships between SOC, TN, SMBC, SMBN and soil properties. Multivariable linear regressions were used to build regression models for SOC, TN, SMBC and SMBN. The GAM was used to test the relationships between SOC, TN and other soil properties in different soil depths. The SEM was conducted to assess the direct or indirect effects of driving factors on SOC, TN, SMBC and SMBN using AMOS version 24.0. The figures were drawn using Origin 9 and Canoco 5 software.

## 5. Conclusions

This study concludes that there were high spatial variabilities for SOC, TN, SMBC and SMBN in the Hongqipao reservoir. Sub-plots which were dominated by *Phragmites australis, Polygonum amphibium* L., *Carex tristachya* and inundated during the study period (C, D and E) had remarkably higher SOC and TN than other sub-plots. This could probably be attributed to a high organic matter supply of the vegetation through litter falling, associated with their relative high productivity. Moreover, inundation could create an unfavorable condition for the mineralization of SOC by soil microbial communities. Sub-plots which were located on dry places within the waterfront zone of the reservoir had relatively low SOC concentrations, probably because mineralization was favored since the soil was dry most of the time. The SOC, TN, SMBC and SMBN contents decreased with increasing soil depth. This was due to the fact that when plant detritus decomposed, most of their organic matters were mineralized and a new soil layer which contained a greater amount of organic carbon was formed. The SOC was significantly positively correlated with TN content in the 0–50 cm soil profile, likely because the vegetation organic matter and litter could be the main nitrogen sources. Moreover, MC was significantly positively correlated with SOC and TN. In contrast to MC, BD was significantly negatively correlated with the SOC and TN contents. No significant correlations were observed between the SOC, TN, SMBC and SMBN contents and soil pH values, and SMB showed weak relationships with other soil properties. The variability in the response variable could be explained better when all of the predicted variables were included in the model. This implied that all of the measured soil variables within the different vegetation types in the reservoir played a crucial role in determining the contents of SOC, TN, SMBC and SMBN. This study further suggests that vegetation types play a major role in determining the spatial characteristics of SOC, TN, SMBC and SMBN. Any changes in the vegetation types in the waterfront area of the reservoir may influence the distributions of SOC, TN, SMBC, SMBN and other soil properties. This may significantly affect the global carbon budget and the atmospheric greenhouse gas concentration.

## Figures and Tables

**Figure 1 plants-12-03767-f001:**
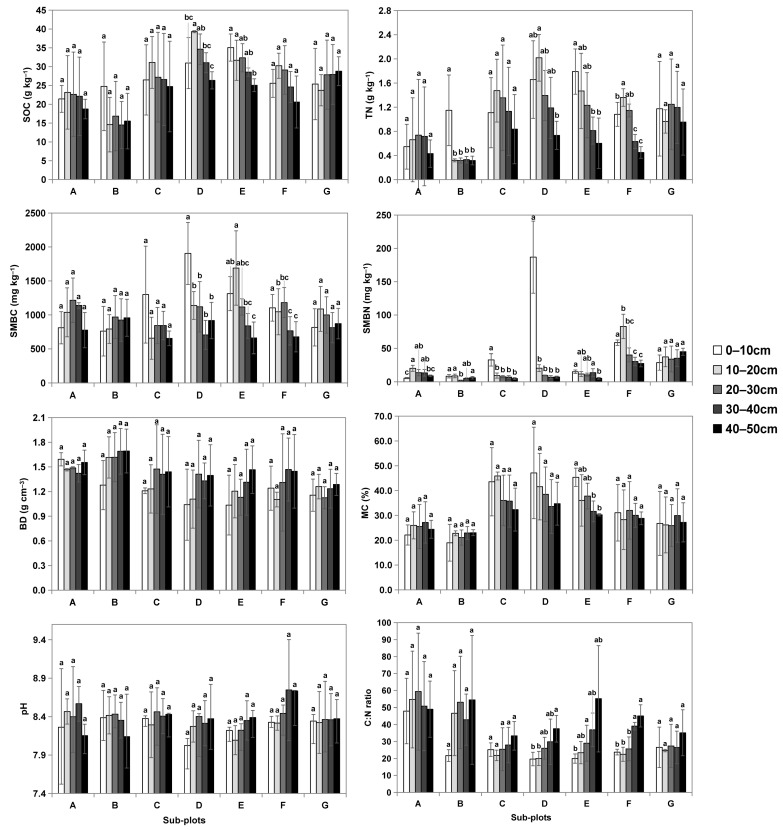
The spatial distributions of soil physicochemical properties in different sub-plots. A–G are seven sub-plots based on different dominant vegetation species and hydrological conditions. Different lowercase letters indicate significant differences between groups, *p* < 0.05.

**Figure 2 plants-12-03767-f002:**
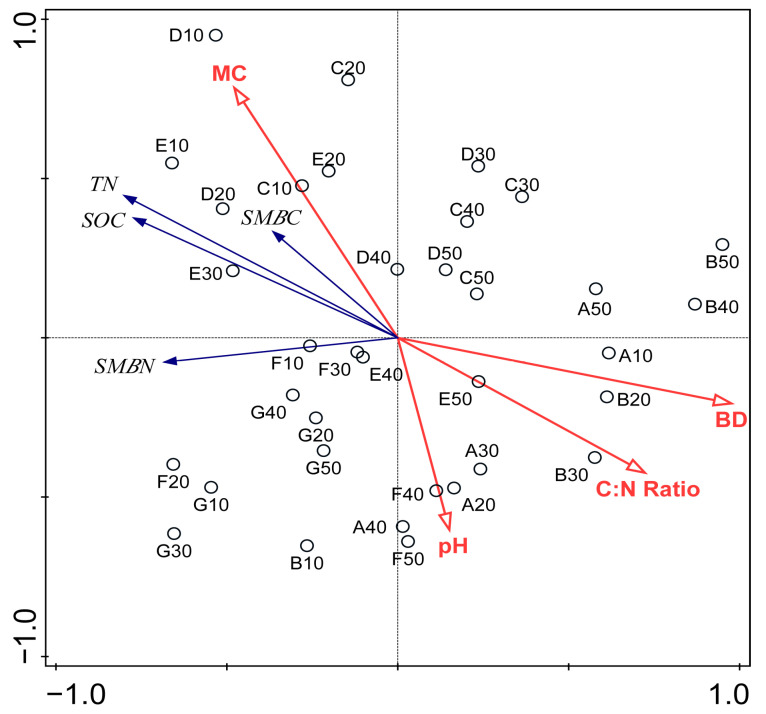
The RDA ordination diagram of the relationships between the SOC, TN, SMBC and SMBN (blue lines) and BD, MC, pH and C:N ratio (red lines). Numbered circles indicate the soil layer in different sub-plots.

**Figure 3 plants-12-03767-f003:**
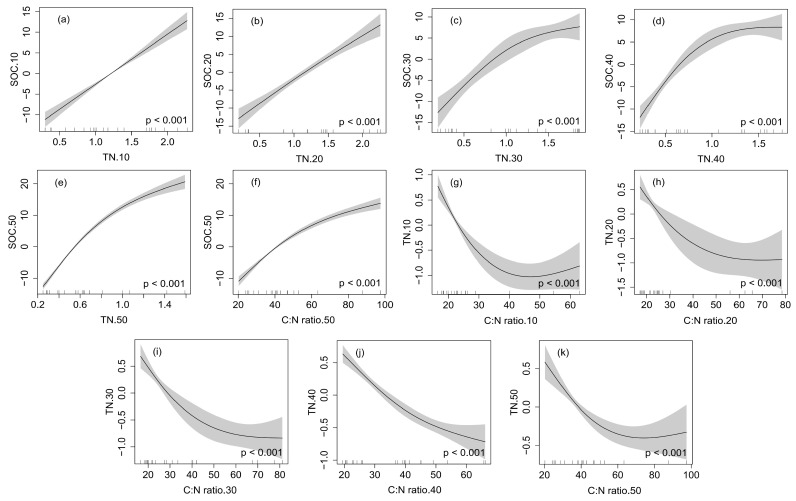
Interrelationships of SOC and TN with other soil physicochemical properties in the 0–50 cm soil layer. (**a**), (**b**), (**c**), (**d**) and (**e**) are the correlations between SOC and TN at 0–10, 10–20, 20–30, 30–40, 40–50 cm soil depths, respectively; (**f**) is the correlation between SOC and C:N ratio at 40–50 cm depth; (**g**), (**h**), (**i**), (**j**) and (**k**) are the correlations between TN and C:N ratio at 0–10, 10–20, 20–30, 30–40, 40–50 cm soil depths, respectively.

**Figure 4 plants-12-03767-f004:**
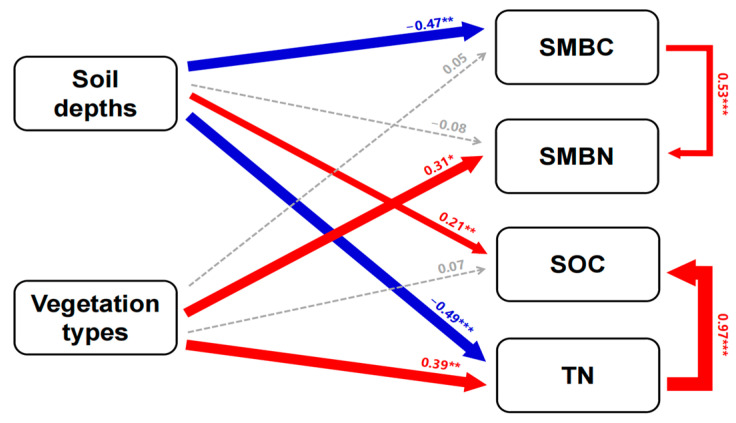
The SEM depicting the effects of vegetation type and soil depth on SOC, TN, SMBC and SMBN (*p* = 0.377, χ^2^ = 1.065, GFI = 0.952, AIC = 37.326, RSMEA = 0.044). The red and blue arrows represent significantly positive and negative pathways, respectively. The gray arrows indicate insignificant pathways. The numbers adjacent to the arrows are standardized path coefficients, analogous to the relative regression weights, and indicative of the effects of the size on the relationships. The thickness of the arrows is proportional to the magnitude of the standardized path coefficient or covariation coefficient. *, ** and *** indicate significance at the 0.05, 0.01 and 0.001 levels, respectively.

**Figure 5 plants-12-03767-f005:**
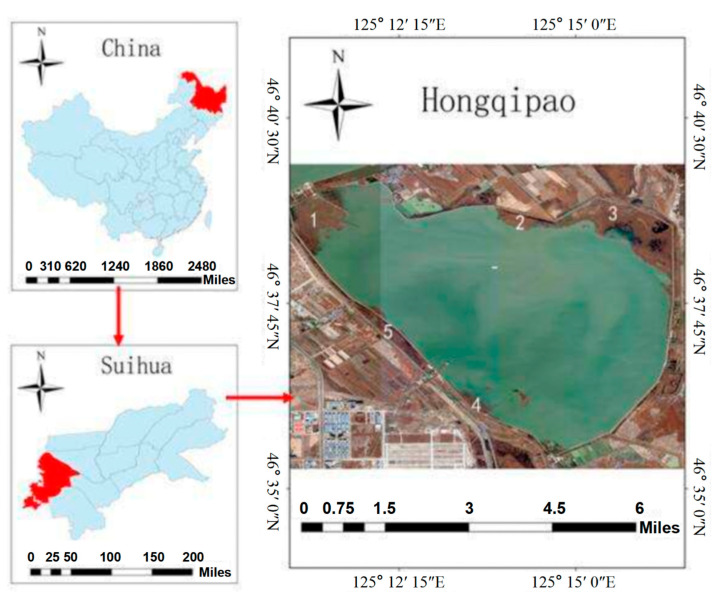
Location of the study area. The top left shows the map of China and the right shows the Hongqipao reservoir. Numbers 1, 2, 3, 4 and 5 refer to the sample plot sites.

**Table 1 plants-12-03767-t001:** Overall mean values of soil properties from different sub-plots in the Hongqipao reservoir.

Soil Properties	A	B	C	D	E	F	G	*p*-Value
SOC (g kg^−1^)	21.62 ± 1.73 cd	17.25 ± 4.31 d	27.22 ± 2.37 ab	32.46 ± 4.81 a	30.54 ± 3.84 ab	26.02 ± 3.84 bc	26.73 ± 2.11 bc	0.000 ***
TN (g kg^−1^)	0.62 ± 0.13 bc	0.48 ± 0.37 b	1.18 ± 0.24 a	1.40 ± 0.49 a	1.18 ± 0.48 a	0.93 ± 0.38 abc	1.10 ± 0.14 ab	0.003 **
C:N ratio	52.38 ± 4.70 a	43.76 ± 13.20 ab	26.76 ± 4.29 d	26.64 ± 7.46 d	32.90 ± 14.00 bc	31.21 ± 10.21 bc	28.07 ± 4.09 d	0.001 ***
SMBC (mg kg^−1^)	995.51 ± 195.82 a	880.71 ± 96.85 a	859.44 ± 262.87 a	1156.67 ± 453.39 a	1123.66 ± 403.70 a	954.79 ± 220.41 a	917.41 ± 121.60 a	0.536
SMBN (mg kg^−1^)	12.44 ± 5.78 a	6.02 ± 2.93 a	12.35 ± 11.59 a	46.22 ± 78.82 a	11.27 ± 3.76 a	47.97 ± 22.97 a	36.14 ± 5.89 ab	0.171
BD (g cm^−3^)	1.50 ± 0.07 ab	1.58 ± 0.17 a	1.35 ± 0.12 bc	1.26 ± 0.17 d	1.23 ± 0.17 d	1.31 ± 0.15 bc	1.21 ± 0.07 d	0.001 ***
MC (%)	25.05 ± 1.92 cd	21.77 ± 1.75 d	38.73 ± 5.73 a	39.11 ± 5.48 a	36.22 ± 5.98 ab	30.06 ± 1.55 bc	27.20 ± 1.62 cd	0.000 ***
pH	8.37 ± 0.17 ab	8.35 ± 0.12 ab	8.39 ± 0.07 ab	8.27 ± 0.15 b	8.25 ± 0.12 b	8.51 ± 0.21 a	8.35 ± 0.02 ab	0.101

** and *** indicate significance at the 0.01 and 0.001 levels, respectively. Different lowercase letters indicate significant differences between groups, *p* < 0.05.

**Table 2 plants-12-03767-t002:** Pearson correlation coefficients between soil physicochemical properties.

	SOC	TN	SMBC	SMBN	C:N Ratio	MC	BD
SOC							
TN	0.91 **						
SMBC	0.35 *	0.46 **					
SMBN	0.25	0.36 *	0.58 **				
C:N Ratio	−0.73 **	−0.86 **	−0.31	−0.37 *			
MC	0.75 **	0.74 **	0.44 **	0.31	−0.61 **		
BD	−0.81 **	−0.86 **	−0.46 **	−0.51 **	0.80 **	−0.60 **	
pH	−0.26	−0.37 *	−0.49 **	−0.28	0.29	−0.28	0.33

* and ** indicate significance at the 0.05 and 0.01 levels, respectively.

**Table 3 plants-12-03767-t003:** Multiple regression models for SOC, TN, SMBC, SMBN and other soil properties.

Y	Factors	Equations	*R* ^2^	*p*
SOC	Correlated	Y = 18.160 + 11.630X_2_ − 0.004 X_3_ + 0.133X_5_ + 0.161X_6_ − 7.264X_7_	0.864	0.000 ***
	All	Y = 0.706 + 11.035X_2_ − 0.002X_3_ − 0.017X_4_ + 0.119X_5_ + 0.161X_6_ − 8.879X_7_ + 2.344X_8_	0.863	0.000 ***
TN	Correlated	Y = 1.840 + 0.040X_1_ + 0.000X_3_ − 0.000X_4_ − 0.013X_5_ + 0.001X_6_− 0.194X_7_ − 0.168X_8_	0.914	0.000 ***
	All	Y = 1.840 + 0.040X_1_ + 0.000X_3_ − 0.000X_4_ − 0.013X_5_ + 0.001X_6_− 0.194X_7_ − 0.168X_8_	0.914	0.000 ***
SMBC	Correlated	Y = 5132.780 − 8.166X_1_ + 161.133X_2_ + 3.394X_4_ + 7.281X_6_ + 62.747X_7_ − 573.348X_8_	0.397	0.002 **
	All	Y = 4428.302 − 18.234X_1_ + 461.896X_2_ + 3.268X_4_ + 10.839X_5_ + 8.099X_6_ − 173.218X_7_ − 466.619X_8_	0.441	0.001 **
SMBN	Correlated	Y = 213.706 − 1.959X_1_ − 18.385X_2_ + 0.056X_3_ − 0.467X_5_ − 117.401X_7_	0.402	0.001 **
	All	Y = 142.587 − 2.565X_1_ − 17.529X_2_ − 0.054X_3_ − 0.395X_5_ + 0.598X_6_ − 124.134X_7_ + 9.076X_8_	0.368	0.005 **

X_1_, SOC; X_2_, TN; X_3_, SMBC; X_4_, SMBN; X_5_, C:N ratio; X_6_, MC; X_7_, BD; X_8_, pH. ** and *** indicate significance at the 0.01 and 0.001 levels, respectively.

**Table 4 plants-12-03767-t004:** Total SOC, TN, SMBC and SMBN storages in different sub-plots.

Sub-Plot Types	TSOC (kg C m^−2^)	TSN (kg N m^−2^)	TSMBC (g C m^−2^)	TSMBN (g N m^−2^)
A	16.29 ± 5.73	0.47 ± 0.46	743.70 ± 140.11	9.26 ± 1.44
B	12.63 ± 4.14	0.35 ± 0.03	705.53 ± 186.67	4.59 ± 0.71
C	17.17 ± 1.09	0.72 ± 0.12	576.20 ± 241.39	7.82 ± 1.47
D	20.17 ± 4.90	0.83 ± 0.06	708.71 ± 268.74	25.72 ± 14.36
E	18.29 ± 3.43	0.69 ± 0.17	680.22 ± 215.21	6.71 ± 1.63
F	16.97 ± 4.30	0.60 ± 0.16	617.62 ± 185.11	30.52 ± 8.60
G	16.10 ± 3.26	0.67 ± 0.29	650.31 ± 169.52	22.43 ± 8.91

**Table 5 plants-12-03767-t005:** Dominant vegetation species, hydrology conditions and aboveground biomass in sub-plots.

Sub-Plots	Dominant Vegetation Species	Hydrology	Biomass (kg m^−^^2^)
A	*Poa annua* L., *Imperata cylindrica*, *Polygonum*	Dry	0.38 ± 0.21
B	*Phragmites australis*, *Mongolian wormwood*	Inundated	0.71 ± 0.03
C	*Tamarix chinensis Lour.*, *Polygonum amphibium* L.	Inundated	3.74 ± 1.73
D	*Scirpus fluviatilis*, *Polygonum amphibium* L., *Carex tristachya*	Inundated	0.55 ± 0.19
E	*Phragmites australis*	Inundated	0.91 ± 0.08
F	*Leymus chinensis*	Dry	0.64 ± 0.17
G	*Poa annua* L.	Dry	0.58 ± 0.01

## Data Availability

All generated data are included in this article.

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
