# Peer review of "Effects of Vegetation Types and Soil Properties on Regional Soil Carbon and Nitrogen in Salinized Reservoir Wetland, Northeast China"

_plants, 2023, doi:10.3390/plants12213767_

Round 1

Reviewer 1 Report

The manuscript entitled “Effects of vegetation types and soil properties on regional soil carbon and nitrogen in salinized reservoir wetland, Northeast China” is a very interesting study. But it has some modifications, as mentioned below:

1.     The charts are not clear such as table 2

2.     The units in Table 5 should be checked.

3.  The manuscript should be checked carefully overall.

4.  The manuscript itself is not well organized.

“The objectives of this study were (iii) to evaluate the total SOC, TN, SMBC and SMBN in the soils collected from sites dominated by different vegetation types in the  Hongqipao reservoir.” How do the authors achieve this objective?

The manuscript entitled “Effects of vegetation types and soil properties on regional soil carbon and nitrogen in salinized reservoir wetland, Northeast China” is a very interesting study. But it has some modifications, as mentioned below:

1.     The charts are not clear such as table 2

2.     The units in Table 5 should be checked.

3.  The manuscript should be checked carefully overall.

4.  The manuscript itself is not well organized.

5 “The objectives of this study were (iii) to evaluate the total SOC, TN, SMBC and SMBN in the soils collected from sites dominated by different vegetation types in the  Hongqipao reservoir.” How do the authors achieve this objective?

Author Response

Dear Reviewer,

According to your advice, the manuscript has been edited. We provide a point-by-point response to your comments. Please see the attachment.

Thank you!

Best regards,

Bing Yu

Reviewer 2 Report

The study examines the effects of vegetation types on soil carbon and nitrogen in Hongqipao reservoir, Northern China. The authors address this by measuring spatial variability in soil organic carbon (SOC), total Nitrogen (TN), soil microbial biomass Carbon (SMBC) and soil microbial biomass Nitrogen (SMBN), all of which were sampled across 7 different sampling sites with varying plant species compositions.

My main concern is the experimental or sampling design, which focused on the description of spatial and depth variation, which considers a sample focused on 7 vegetation formations. This was done by carrying out one way ANOVA analysis, with sites as the factor. This is then complemented by correlation an multiple linear regression analysis, as well as RDA.

However, Figure 1 shows the data organised or grouped by soil depths and sites, which seems to imply a two-way analysis of variance. It seems that there are not enough replicates to do this, and hence the study design may not provide adequate responses to the overall question. If the choice is to use different depths as pseudo replicates for the site comparison, then it would be necessary to show that there are no significant differences across different depts. While the authors indicate in lines 122 to 127 that some response variables change with depth, no statistical analyses are shown. 

In this regard, given the sample size, and study design, the overarching question, namely, how do different vegetation types affect soil carbon and nitrogen properties, can only be tackled in a descriptive manner. In addition, the one-way ANOVA detects significant differences among study sites. However, no post hoc tests are shown, which would allow a better understanding of which are the sites that differ in the response variables. In addition, while the authors examine the Pearson correlation coefficients in Table 3, Figure 3 actually shows that some of these relationships seem to be nonlinear. However, no statistical analysis is shown for Figure 3, hence there is no way to assess whether the shown interrelationships are significant or not, and what is the role of depth (or site) across these data sets. 

Overall, I would encourage you to clarify the statistical analysis, showing whether the effect of depth is significant, and including those factors in a multiple linear regression or a lineal model. Should those not be amenable due to sample size, I would suggest the use of a classification and regression tree (CART) or other decision tree techniques to approach the understanding of how variation in SOC, TN,SMBC and SMBN reflects either spatial variation in vegetation composition. 

A few final questions that remain is where are the sampling sites located, relative to the Hongqipao reservoir. I would suggest that they be shown in Figure 4. In addition, it is not clear if the variation in vegetation could be assumed to be reflected in vegetation types or formations, or whether the distribution is actually something that does not form clear cut vegetation communities. This information seems relevant in the light of the overall research question. 

In addition to these main concerns, I found some minor typos which I indicate below:

Line 20 : I suggest  " attributed by the fact" be edited to " attributed to the fact"

Line 24: "liter" should be editted to "litter"

Line 25: " positive" should be edited to "positively"

Line 49: Please check the formatting of the number " 5 ×105 km2". It would seem that there's a missing superscript for "105'.  Also note that there are other, up to date available sources for this information, such as RealSAT database (https://doi.org/10.1038/s41597-022-01449-5), or other up to date databases (https://doi.org/10.1038/s41598-022-17074-6, https://doi.org/10.1038/s41597-022-01425-z)

Author Response

(The authors gave the same response as above.)
